# Regulating Oral Biofilm from Cariogenic State to Non-Cariogenic State via Novel Combination of Bioactive Therapeutic Composite and Gene-Knockout

**DOI:** 10.3390/microorganisms8091410

**Published:** 2020-09-13

**Authors:** Hong Chen, Yingming Yang, Michael D. Weir, Quan Dai, Lei Lei, Negar Homayounfar, Thomas W. Oates, Kai Yang, Ke Zhang, Tao Hu, Hockin H. K. Xu

**Affiliations:** 1State Key Laboratory of Oral Diseases, Department of Operative Dentistry and Endodontics, West China School of Stomatology, National Clinical Research Centre for Oral Diseases, Sichuan University, Chengdu 610041, China; 501411@cqmu.edu.cn (H.C.); ymyang@scu.edu.cn (Y.Y.); leilei@scu.edu.cn (L.L.); 2Department of Conservative Dentistry and Endodontics, College of Stomatological, Chongqing Medical University, Chongqing Key Laboratory of Oral Diseases and Biomedical Sciences, Chongqing Municipal Key Laboratory of Oral Biomedical Engineering of Higher Education, Chongqing 401147, China; 3Department of Advanced Oral Sciences and Therapeutics, University of Maryland Dental School, Baltimore, MD 21201, USA; MWeir@umaryland.edu (M.D.W.); qdai@umaryland.edu (Q.D.); nhomayounfar@umaryland.edu (N.H.); toates@umaryland.edu (T.W.O.); 4Clinical Research Center of Shanxi Province for Dental and Maxillofacial Diseases, Key Laboratory of Shaanxi Province for Craniofacial Precision Medicine Research; College of Stomatology, Xi’an Jiaotong University, Xi’an 710004, China; 5Department of Orthodontics, School of Stomatology, Capital Medical University, Beijing 100069, China; dr_yangkai@163.com; 6Center for Stem Cell Biology & Regenerative Medicine, University of Maryland School of Medicine, Baltimore, MD 21201, USA; 7Marlene and Stewart Greenebaum Cancer Center, University of Maryland School of Medicine, Baltimore, MD 21201, USA

**Keywords:** multi-species biofilm, *S. mutans**rnc* gene-knockout, bioactive composite, regulating biofilm composition, biofilm–material interactions, dental caries

## Abstract

The objectives were to investigate a novel combination of gene-knockout with antimicrobial dimethylaminohexadecyl methacrylate (DMAHDM) composite in regulating oral biofilm from a cariogenic state toward a non-cariogenic state. A tri-species biofilm model included cariogenic *Streptococcus mutans* (*S. mutans*), and non-cariogenic *Streptococcus sanguinis* (*S. sanguinis*) and *Streptococcus gordonii* (*S. gordonii*). Biofilm colony-forming-units (CFUs), lactic acid and polysaccharide production were measured. TaqMan real-time-polymerase-chain reaction was used to determine the percentage of each species in biofilm. The *rnc* gene-knockout for *S. mutans* with DMAHDM composite reduced biofilm CFU by five logs, compared to control (*p* < 0.05). Using parent *S. mutans*, an overwhelming *S. mutans* percentage of 68.99% and 69.00% existed in biofilms on commercial composite and 0% DMAHDM composite, respectively. In sharp contrast, with a combination of *S. mutans rnc* knockout and DMAHDM composite, the cariogenic *S. mutans* percentage in biofilm was reduced to only 6.33%. Meanwhile, the non-cariogenic *S. sanguinis* + *S. gordonii* percentage was increased to 93.67%. Therefore, combining *rnc*-knockout with bioactive and therapeutic dental composite achieved the greatest reduction in *S. mutans*, and the greatest increase in non-cariogenic species, thereby yielding the least lactic acid-production. This novel method is promising to obtain wide applications to regulate biofilms and inhibit dental caries.

## 1. Introduction

To date, over 700 phylotypes have been found in the oral cavity of healthy people and people with diseases [1]. Even healthy people can present cariogenic bacteria [2,3]. It is important to maintain the balance of oral microbiome for oral and body health [4,5]. Dental caries is one of the most widespread and costly biofilm-mediated oral infectious diseases, affecting all ages of people worldwide [6,7]. Dental caries is associated with dysbiosis of the tooth-colonizing microbiota, with the species types in the dental plaque biofilm shifting from a healthy composition toward a cariogenic composition [8,9]. This causes a microbial shift of dental plaque, increasing the growth and metabolism of cariogenic bacteria while inhibiting the beneficial organisms [10]. Oral streptococci are the early colonizers, and their initial adhesion determines the species composition in the oral biofilm and impacts on the health status of the host [11]. Interspecies interactions of oral streptococci play an important role in the biofilm shift [12]. Previous studies established a three oral streptococci species model: cariogenic *Streptococcus mutans* (*S. mutans*), and noncariogenic *Streptococcus sanguinis* (*S. sanguinis*) and *Streptococcus gordonii* (*S. gordonii*) [12]. This three-species biofilm model is referred to as the Sm + Ss + Sg biofilm model in the present paper. These three species compete for adhesion-binding sites on the tooth surface, and compete for carbohydrates [13]. Among them, *S. mutans* can reduce plaque pH upon the ingestion of fermentable carbohydrate, and induce the selection for a cariogenic flora at the expense of the healthy and less aciduric residents, leading to lower proportions of *S. sanguinis* and *S. gordonii* in the biofilm [14,15]. On the other hand, *S. sanguinis* and *S. gordonii* can generate H_2_O_2_ to decrease the growth of *S. mutans* [12,14].

Resin composites are popular dental materials to restore tooth cavities. However, compared with amalgam and glass ionomers, resin composites tend to collect more biofilms [16,17]. Biofilm acids can lead to tooth mineral dissolution, causing tooth decay. Therefore, secondary caries (recurrent caries) is one of the major reasons for the failure of tooth cavity restorations [18,19,20]. As a result, there is an urgent need to develop a new generation of bioactive and therapeutic dental resin composites to control oral biofilms and inhibit tooth decay.

Quaternary ammonium methacrylates (QAMs) are cationic and can be co-polymerized in dental resin to obtain long-term contact-inhibition against bacteria [21,22,23]. In previous studies, several novel QAMs were used in a dental product, including 12-methacryloyloxydodecyl-pyridinium bromide (MDPB) [24], quaternary ammonium dimethacrylate (QADM) [25], quaternary ammonium-polyethylenimine (QPEI) [26,27], methacryloxylethyl cetyl dimethyl ammonium chloride (DMAE-CB) [28], dimethylaminododecyl methacrylate (DMADDM) [29] and dimethylaminohexadecyl methacrylate (DMAHDM) [30]. In addition, these QAMs with varying alkyl chain lengths had acceptable low cytotoxicity against human gingival fibroblasts and odontoblast-like cells, matching the low cytotoxicity of commercial dental monomers in clinical use [31,32]. DMAHDM with an alkyl chain length of 16 was incorporated into resins and presented a strong antimicrobial ability with a low level of cytotoxicity [30,32]. Several studies showed the ability of QAMs to modulate oral biofilm composition to promote noncariogenic species and suppress cariogenic species [7,8].

As we have recently learned, the genes and pathways of *S. mutans* play an important role in the interactions with anti-caries compounds [33,34]. The *rnc* gene is a post-transcriptional regulator gene in *S. mutans*, and previous studies proved a knockout of *rnc* gene in *S. mutans* could suppress acid production and inhibit caries [35,36]. However, these studies tested a single species biofilm, and did not test biofilm growth and species modulation on composite surfaces. To date, there has been no report on the effects of *rnc* knockout in *S. mutans* on the modulation of bacterial species in the Sm + Ss + Sg biofilm; it is unknown whether *rnc* knockout would shift the biofilm composition from a cariogenic state toward a non-cariogenic healthy composition. Furthermore, there has been no report on the effects of combining *rnc* knockout in *S. mutans* with DMAHDM composite on the modulation of the Sm + Ss + Sg biofilm.

Therefore, the objectives of this study were to investigate, for the first time: (1) the effect of *rnc* gene knockout in *S. mutans* on species proportional shift in a tri-species biofilm model (*S. mutans*, *S. sanguinis,* and *S. gordonii*); (2) the dual strategy of combining the *rnc* gene-knockout for *S. mutans* with a bioactive and therapeutic composite in regulating the biofilm to shift from a cariogenic state toward a non-cariogenic state. It was hypothesized that: (1) The *S. mutans rnc* gene-knockout would substantially reduce the cariogenic *S. mutans* percentage in the Sm + Ss + Sg biofilm and increase the non-cariogenic species percentage, even without the use of an antimicrobial agent; (2) DMAHDM composite would induce a shift in biofilm species from a cariogenic composition toward a non-cariogenic composition; (3) The dual strategy of combining the *rnc* gene-knockout with DMAHDM composite would achieve the greatest reduction in *S. mutans* percentage in the Sm + Ss + Sg biofilm, and obtain the greatest increase in the non-cariogenic species percentage, thereby resulting in the least acid secretion by the biofilm.

## 2. Materials and Methods

### 2.1. Composites Fabrication

DMAHDM was produced using a modified Menschutkin reaction technique [37]. The main merit of this reaction is that the products need minimal purification before experimental use. Briefly, 10 mmol of 2-(dimethylamino) ethyl methacrylate (Sigma-Aldrich, St. Louis, MO, USA), 10 mmol of 1-bromododecane (TCI America, Portland, OR, USA), and 3 g of ethanol were mixed and then agitated at 70 °C for 24 h. The ethanol solvent was eliminated via evaporation and DMAHDM was then obtained [38].

Heliomolar (Ivoclar, Amherst, NY, USA) served as a commercial control composite. According to the manufacturer, bisphenol A glycidyl dimethacrylate (BisGMA, Esstech, Essington, PA, USA) and triethylene glycol dimethacrylate (TEGDMA, Esstech) were mixed at 1:1 mass ratio (all by mass unless otherwise noted) [23,39]. Then, 0.2% camphorquinone and 0.8% ethyl 4-*N*,*N*-dimethylaminobenzoate were incorporated for photoactivation; this resin is termed BT resin [35]. DMAHDM/(BT + DMAHDM) mass fractions of 0%, 5%, and 10% were used. The highest concentration of 10% was chosen without compromising the mechanical property [8]. Barium boroaluminosilicate glass particles with a median size of 1.4 μm (Caulk/Dentsply, Milford, DE, USA) were silanized with 4% 3-methacryloxypropyltrimethoxysilane [35,40]. A mass fraction of 70% of glass particles was mixed into BT resin, and the DMAHDM mass fractions in the composite were 0%, 1.5%, and 3%, respectively [8]. Thus, the following four groups were investigated:(1)Commercial control: Heliomolar;(2)0% DMAHDM: 30% BT + 0% DMAHDM + 70% glass particles;(3)1.5% DMAHDM: 28.5% BT + 1.5% DMAHDM + 70% glass particles;(4)3% DMAHDM: 27% BT + 3% DMAHDM + 70% glass particles.

To make resin disks, the molds with a diameter of 10 mm and a thickness of 2 mm was used, and then light-cured (Triad 2000; Dentsply, York, PA, USA) for 1 min on each side were applied [8]. The cured disks were submerged in 200 mL of distilled water and agitated magnetically with a bar at a speed of 100 r /min for 24 h [8]. Then the disks were sterilized with ethylene oxide (AnproleneAN 74i, Andersen, Haw River, NC, USA) and degassed for 3 days [23].

### 2.2. Mechanical Testing

Six specimens of 2 × 2 × 25 mm dimensions were tested for each group (*n* = 6) for load-bearing properties [39]. The specimens were photo-polymerized (Triad 2000, Dentsply, York, PA, USA) for 1 min on each side, and then incubated for 24 h at 37 °C. Flexural strength and elastic modulus were investigated via three-point flexure employing a 10 mm span at a crosshead speed of 1 mm/min on a computer-controlled Universal Testing Machine (5500R, MTS, Cary, NC, USA) [40]. Flexural strength was calculated by the equation of *S* = 3*P_max_*/*L*(2*bh*^2^), where *P_max_* = the fracture load, *L* = the span, *b* = specimen width, and *h* = specimen thickness [41]. Elastic modulus was determined by the equation of *E* = (*P*/*d*)(*L*^3^/[4*bh*^3^]), where load *P* divided by the related displacement *d* is the curve’s slope in the linear elastic region [41].

### 2.3. Bacterial Strains and Growth Conditions

The use of all the bacterial species was approved by the University of Maryland Baltimore Institutional Review Board. Parent *S. mutans* strain UA159 (ATCC 700610) and the *rnc*-knockout strain were provided by West China School of Stomatology [35,36]. Prior to the experiments, the accuracy of the strains was verified using polymerase chain reaction (PCR) and sequencing [35]. *Streptococcus sanguinis* ATCC10556 (*S. sanguinis*) and *Streptococcus gordonii* ATCC10558 (*S. gordonii*) were obtained from American Type Culture Collection (ATCC, Manassas, VA, USA). All the bacterial strains are listed in Table 1 [35]. They were cultured in a brain-heart infusion (BHI) broth (Sigma, St. Louis, MO, USA) at 37 °C with 5% CO_2_. For Sm + Ss + Sg biofilm formation, the sterilized composite disks were placed into 24-well plates. Then, the overnight parent or *rnc*-knockout *S. mutans*, and *S. sanguinis*, *S. gordonii* were adjusted to 10^7^ colony-forming units (CFUs)/mL in 1.5 mL BHI supplied with 1% sucrose for each well [8]. For 2-day biofilms, the composite disks with biofilms were transferred to new 24-well plates filled with fresh culture medium without the washing steps after 24 h. Then the cultures were incubated at 37 °C with 5% CO_2_ for another 24 h. The 2-day biofilms were then used for subsequent experiments.

### 2.4. Live/Dead Staining Assay

Disk samples grown with 2-day biofilms were washed with phosphate buffered saline (pH 7.4) (PBS). Live/Dead Baclight bacterial viability kits (Molecular Probes, Eugene, OR, USA) were used as reported in previous investigations [8,35]. The 2.5 μM SYTO 9 and 2.5 μM propidium iodide were used to label the live and dead bacterial cells, respectively. Images of five random fields of each group were captured on each disk, and three disks were used for each group, thus producing 15 images for each group.

### 2.5. Colony-Forming Unit (CFU) Counts

Biofilms grown for two days on composite disks were transferred into tubes with 2 mL of PBS, and the biofilms were collected by employing a sonication and vortex method (Fisher, Pittsburg, PA, USA) [42]. The bacterial suspensions were successively diluted and then moved to BHI agar plates. The agar plates were cultured at 37 °C in 5% CO_2_ for 48 h, and the CFU counts were calculated by counting the colony numbers.

### 2.6. Biofilm Viability Using the MTT Assay

The 3-(4,5-dimethylthiazol-2-yl)-2,5-diphenyl tetrazolium bromide (MTT) (VWR Chemicals, OH, USA) assay was used to estimate the viability of bacteria in biofilms [8]. Disks with 2-day biofilms were washed twice with PBS and transferred into a new 24-well plate (*n* = 6). 1 mL MTT dye (0.5 mg/mL MTT in PBS) was added into each well and incubated at 37 °C in 5% CO_2_ for 1 h [8,29]. The disks were then transferred to a new 24-well plate, and 1 mL dimethyl sulfoxide (DMSO) was added in each well at room temperature for 20 min to dissolve the formazan crystals. After mixing via pipetting, 200 μL of the DMSO solution was collected and transferred into 96-well plate. OD540 nm was determined via the microplate reader (SpectraMax M5, Molecular Devices, Sunnyvale, CA, USA) [8].

### 2.7. Polysaccharide Synthesis in Biofilms

The water-insoluble polysaccharides production by 2-day biofilms was evaluated using a phenol-sulfuric acid method [8,40,43]. Biofilms were immersed in 2 mL PBS collected by sonication/vortexing. Centrifugation yielded a precipitate, which was washed twice with PBS. The precipitate was then resuspended in 100 μL of distilled water. 100 μL of 6% phenol solution and 0.5 mL of 95–97% sulfuric acid was added, followed by incubation for 30 min. Then, 200 μL of the solution was transferred into a 96-well plate and OD_490nm_ was determined with the microplate reader (SpectraMax M5, Molecular Devices). Six glucose concentrations of 0, 5, 10, 20, 50 and 100 mg/mL were used to plot the standard curve of OD_490nm_ readings to polysaccharide concentrations.

### 2.8. Lactic Acid Secretion by Biofilms

Disks with 2-day biofilms were rinsed twice with PBS and removed to a new 24-well plate. They were immersed in 1.5 mL buffered peptone water (BPW, Sigma-Aldrich) with 0.2% sucrose and incubated at 37 °C in 5% CO_2_ for 3 h (*n* = 6) [25]. The lactate concentrations in BPW were determined using an enzymatic (lactate dehydrogenase) method by measuring OD_340nm_, and the content was determined by the standard curves [29].

### 2.9. DNA Isolation and TaqMan Real-Time PCR Assay

TaqMan real-time PCR was used to quantify the bacteria composition shifts at 2-day biofilms using a Premix Ex Taq (Probe qPCR). The species-specific primers and probes used for *S. mutans*, *S. sanguinis* and *S. gordonii* were designed following the methods described previously (Table 2) [8]. Conventional PCR was used to confirm the specificity of the primers. Biomass samples were collected, and total bacterial DNA isolation and purification were performed with the Genomic DNA Extraction Kit (TaKaRa Biotechnology Co., Ltd., Japan) following the manufacturer’s protocol [8]. NanoDrop 2000 spectrophotometer (Thermo Scientific, Waltham, MA, USA) was used to determine the purity and concentration of the DNA. The absolute quantification was determined using Quant-Studio 3D Digital PCR System (Thermo Fisher Scientific) and analyzed with QuantStudio 3D AnalysisSuite Cloud Software (Thermo Fisher Scientific). The standard curves for quantification of the three bacterial species were generated by 10-fold dilutions of DNA extracted from the bacteria at 10^8^ CFUs to 10^3^ CFUs versus the logarithm of the concentration [44]. Based on the standard curves, the numbers of each strain on the disks were determined.

### 2.10. Statistical Analysis

Statistical analyses were performed using Statistical Package for the Social Sciences (SPSS 22.0, Chicago, IL, USA). All data were expressed as the mean value ± standard deviation (mean ± SD). One-way analyses of variance (ANOVA) were performed to detect the statistical significance of the variables. Tukey’s multiple comparison test was used to compare the means of each of the groups. The differences in the means of data were considered significant if *p* < 0.05.

## 3. Results

The load-bearing properties of the four composites (mean ± SD; *n* = 6) are presented in Figure 1. Compared to commercial control composite, the experimental composite control had significantly higher flexural strength and elastic modulus (*p* < 0.05), and the load-bearing capabilities were not negatively affected with the different mass fractions of DMAHDM (*p* > 0.1).

Representative 2-day mature biofilm live/dead images are shown in Figure 2. The commercial control composite had mostly green staining of live bacteria. Composites with 0% DMAHDM had similar images to commercial control. With the DMAHDM content increasing to 1.5% (E, F), and 3% (G, H), the amount of red staining of the compromised bacteria increased in the biofilms.

As shown in Figure 3A, the Sm + Ss + Sg biofilm CFU counts of biofilms cultured for 2 days (mean ± SD; *n* = 6) on DMAHDM composites were reduced with higher DMAHDM mass fraction (*p* < 0.05). Combining *rnc* knockout for *S. mutans* with 3% DMAHDM composite achieved the greatest biofilm inhibition. Compared to the control group, the CFUs of the *rnc* gene-knockout biofilms in 1.5% and 3% DMAHDM groups was reduced by about three logs and five logs, respectively (*p* < 0.05). Similarly, Figure 3B indicates that combining *rnc* gene-knockout for *S. mutans* with DMAHDM composite led to the least biofilm metabolic activity (*p* < 0.05).

The standard curve of OD_490nm_ against the polysaccharide content was expressed as y = 0.0207x + 0.3082, and the coefficient of determination (R^2^) was 0.9836. A linear curve for the standard lactic acid (Supelco Analytical, Bellefonte, PA, USA) was obtained within the concentration range of 0–3.5 mM. The equation of the regression was y = 8.467x − 1.973, and the coefficient of determination (R^2^) was 0.9942. In Figure 4A, the polysaccharide production of the 2-day Sm + Ss + Sg biofilms on composites are plotted (mean ± SD; *n* = 6). Biofilm polysaccharide amount was greatly reduced with increasing DMAHDM mass fraction, compared to control composites (*p* < 0.05). In addition, the decreasing trend was more profound in biofilms with *rnc*-knockout *S. mutans* than that of parent *S. mutans* (*p* < 0.05). The lactic acid secretion (Figure 4B) was decreased with increasing DMAHDM mass fraction (*p* < 0.05). The combination of *rnc*-knockout in *S. mutans* with the 3% DMAHDM composite led to the biggest decrease in lactic acid secretion by the Sm + Ss + Sg biofilms.

Standard curves of C_t_ value versus bacterial numbers of *S. mutans*, *S. sanguinis* and *S. gordonii* were constructed. The equation of the regression for *S. mutans* was y = −3.3043x + 43.794, and the coefficient of determination (R^2^) was 0.9997. The equation of the regression for *S. sanguinis* was y = −3.2962x + 48.733, and the coefficient of determination (R^2^) was 0.9964. The equation of the regression for *S. gordonii* was y = −3.3246x + 48.305, and the coefficient of determination (R^2^) was 0.9988.

The TaqMan real-time PCR results demonstrated the ratio variation of bacteria in 2-day multispecies biofilms with different DMAHDM mass fractions (Figure 5) (mean ± SD; *n* = 3). With parent *S. mutans*, an overwhelming *S. mutans* proportion of 68.99% and 69.00% existed in the Sm + Ss + Sg biofilm on commercial control composite and 0% DMAHDM, respectively. In contrast, the patent *S. mutans* proportion decreased with increasing the mass fraction of DMAHDM, accounting for 36.67% in 1.5% DMHADM group, and 13.56% in 3% DMHADM group.

Combining *rnc*-knockout for *S. mutans* with DMAHDM composite led to the biggest reduction in *S. mutans* percentage in the Sm + Ss + Sg biofilm. The fraction of *rnc*-knockout *S. mutans* decreased to 16.33% in the 1.5% DMHADM composite group, and to only 6.33% in the 3% DMHADM composite group.

Figure 6 shows the influence of *rnc*-knockout for *S. mutans* alone, DMAHDM composite alone, and *rnc*-knockout for *S. mutans* plus DMAHDM composite, on bacterial composition shift of Sm + Ss + Sg biofilm: (A) The cariogenic *S. mutans* proportion, (B) non-cariogenic *S. sanguinis* proportion, and (C) lactic acid secretion (mean ± SD; *n* = 6). These results demonstrate that: (1) The *rnc*-knockout for *S. mutans* regulated the Sm + Ss + Sg biofilm species composition shift toward a healthier state and decreased the lactic acid secretion, even on composites without DMAHDM; (2) DMAHDM composite had the ability to regulate biofilm equilibrium and reduce the lactic acid secretion; (3) The combination of *rnc*-knockout in *S. mutans* with DMAHDM composite led to the biggest reduction in cariogenic *S. mutans* percentage in Sm + Ss + Sg biofilm, gained the most substantial increase in non-cariogenic species percentage, yielding the least lactic acid secretion.

## 4. Discussion

This present study represents the first report on the effect of *rnc* gene knockout in *S. mutans* on species proportional shift in a tri-species biofilm model, and furthermore, on the effect of combining *S. mutans rnc* knockout with DMAHDM composite in regulating the biofilm from a cariogenic state toward a non-cariogenic state. The *S. mutans rnc* gene-knockout alone significantly decreased the cariogenic *S. mutans* percentage in the Sm + Ss + Sg biofilm and increased the non-cariogenic species percentage. The DMAHDM composite alone caused a shift in biofilm species from a cariogenic composition toward a non-cariogenic composition. Most dramatically, by combining the *rnc* gene-knockout technology with the antimicrobial DMAHDM composite, the *S. mutans* percentage in the Sm + Ss + Sg biofilm was decreased from an overwhelming 69% of the parent *S. mutans* on control composite to only 6.3% of *rnc*-knockout *S. mutans* on the 3% DMAHDM composite. Meanwhile, the corresponding non-cariogenic *S. sanguinis* proportion in the Sm + Ss + Sg biofilm was increased from 12% to 61%. Therefore, the dual strategy of *rnc*-knockout for *S. mutans* + DMAHDM composite is promising to be a viable method for regulating the biofilm composition to promote non-cariogenic species and suppress cariogenic species, thereby inhibiting biofilm acids and recurrent caries.

QAMs are promising for incorporation into composites, primers and adhesives due to their antimicrobial activity and low toxicity [32,47,48]. DMAHDM has a positively-charged quaternary amine N^+^ and a long alkyl chain with a chain length of 16 [49,50]. N^+^ can contact the negatively-charged cell membrane of bacteria and alter the electric balance, leading the bacteria to explode under its own osmotic pressure [49,50]. Meanwhile, the long cationic polymers can penetrate bacterial cells by disrupting the membranes [8,49]. DMAHDM can be co-polymerized and covalently bonded in resins to provide long-term contact-inhibition against bacteria [49,50]. This study showed that the incorporation of 3% DMAHDM into the composite did not negatively affect the flexural strength and the elastic modulus as compared to those of commercial control composite and 0% DMAHDM.

Interactions within oral microbial communities are major factors affecting the development of biofilm, and the cariogenic potential of the oral microbial consortia depends on the environmental conditions and the composition of the bacterial flora [44,51]. For a better representation of oral microorganisms, an Sm + Ss + Sg biofilm model was used to include *S. mutans, S. sanguinis* and *S. gordonii* [7,8,44]. As a key modulator, *S. mutans* is the primary exopolysaccharides (EPS) producer in the oral cavity and is responsible for dental caries [14,15]. The acidification of the EPS-rich matrix favors the growth of the cariogenic pathogens, promotes demineralization of the enamel apatite on the teeth, and leads to the onset of dental caries [52,53]. Although *S. mutans* is an important player in caries pathogenesis, its role in caries is not etiological [54]. The correlation between caries experience with elevated numbers of *S. mutans* has been controversial and not well-established [54,55,56]. Some studies demonstrated a real association between *S. mutans* and the carious lesion, whereas other studies revealed no clear association [57]. In some studies, caries lesions occurred without the presence of *S. mutans* [58].

Oral streptococci account for 20% of all supragingival microorganisms, but they account for nearly 80% of the initial colonizers during the early stages of biofilm formation [56]. Furthermore, oral streptococci have been associated with the onset and progression of carious lesions [8,56]. When being present in the oral biofilm in high numbers, the pioneer colonizers *S. sanguinis* and *S. gordonii* can antagonize *S. mutans* [59]. According to clinical studies, a great percentage of *S. mutans* was always found in the caries lesion, which indicated its acidogenicity and cariogenicity [8,60,61]. In contrast, higher percentages of *S. sanguinis* and *S. gordonii* in dental plaque were associated with sound tooth surfaces without caries [8,60,61].

The exopolysaccharides (EPS) matrix is a key virulence factor. It serves as nutrients and helps form the core of the matrix scaffold [52]. In addition, the EPS matrix does contribute towards drug resistance which can be a diffusion barrier to stop the antimicrobial agents [53]. Previous studies indicated that targeting the biofilm EPS matrix may be an effective strategy for removing biofilms, killing bacteria and disrupting the pathogenic environment [62]. In the present study, the *rnc* knockout for *S. mutans* disrupted the polysaccharide synthesis in the Sm + Ss + Sg biofilm. The fewer polysaccharides indicated an easier exposure to antimicrobial agents [63]. Therefore, the *rnc* knockout for *S. mutans* reduced the virulence factor and rendered the biofilm to become more sensitive to DMAHDM composite. This synergistic effect from the combination of *rnc*-knockout for *S. mutans* with DMAHDM composite achieved the greatest biofilm-inhibition efficacy, reducing biofilm CFUs by five logs.

Biofilms can produce organic acids, lower the pH, and induce the selection for an acidogenic and aciduric flora, thus shifting the biofilm composition from a healthy state to a cariogenic state [8]. In the present study, the lactic acid secretion by biofilms decreased significantly on DMAHDM-containing composite, especially at the mass fraction of 3%. Moreover, the combination *rnc*-knockout for *S. mutans* with the 3% DMAHDM composite achieved the greatest reduction in the lactic acid secretion of the Sm + Ss + Sg biofilm by an order of magnitude. The substantial reduction in acid secretion helped the non-cariogenic species to survive and grow, thus maintaining a healthy non-cariogenic biofilm composition.

Indeed, the combined contribution of *rnc* knockout in *S. mutans* and DMAHDM composite on biofilm modulation was remarkable. With only 6.3% *S. mutans* in the biofilm, the non-cariogenic species can grow and thrive. Therefore, *rnc* knockout in *S. mutans* plus DMAHDM composite demonstrated the ability to bring the biofilm species balance back toward a healthy state for the first time. This percentage shift of bacteria was attributed to the dual pressure of both the competition among bacteria and the antimicrobial agent. This was because: (1) the antimicrobial effect from DMAHDM exerted an acid-inhibition effect to weaken the competitiveness of *S. mutans* in the Sm + Ss + Sg biofilm; (2) *rnc* knockout for *S. mutans* produced less polysaccharide in Sm + Ss + Sg biofilm models which decreased the biofilm drug-tolerance and induced greater sensitivity to DMAHDM; (3) *S. mutans rnc*-knockout plus DMAHDM composite achieved the greatest reduction in lactic acid secretion, which helped the non-cariogenic and non-aciduric species to survive and grow [8,12]. Therefore, the use of *rnc* gene-knockout in *S. mutans* plus the bioactive and therapeutic composite caused the greatest decrease in biofilm acid secretion, which provided an environment in favor of the non-cariogenic species, which would be clinically beneficial to prevent caries. A previous study [64] reported a unique method of probiotic bacterial colonization to prevent tooth decay. In addition, further study is still needed to investigate the *S. mutans rnc* gene-knockout effect in an oral biofilm model in vivo with animal experiments. Further studies are needed to investigate the dual strategy of *rnc* gene-knockout in *S. mutans* plus DMAHDM in regulating biofilm development tendency to a healthy composition by testing more species to mimic the complexity of oral biofilms.

## 5. Conclusions

This study demonstrated the high efficacy of *rnc* gene knockout in *S. mutans* and a bioactive and therapeutic composite in Sm + Ss + Sg biofilm modulation to suppress the cariogenic species percentage and increase the non-cariogenic species percentage for the first time. The *S. mutans rnc* gene-knockout alone reduced the cariogenic *S. mutans* percentage in the Sm + Ss + Sg biofilm. The DMAHDM composite induced a shift in biofilm species from a cariogenic composition toward a non-cariogenic composition. The dual strategy of combining *rnc* gene-knockout with DMAHDM composite yielded the greatest reduction in *S. mutans* percentage in the Sm + Ss + Sg biofilm, and the greatest increase in the non-cariogenic species percentage, thereby resulting in the lowest amount of biofilm acid. The dual strategy reduced the cariogenic *S. mutans* proportion from 69% on commercial composite, to 6% on 3% DMAHDM. Combining the *rnc* knockout for *S. mutans* with DMAHDM composite reduced the biofilm CFUs by five logs. Therefore, the novel dual strategy of bacterial gene-modification and bioactive and therapeutic dental resin has the potential to modulate biofilm shift from a cariogenic state to a non-cariogenic state. This novel dual strategy is promising for a wide range of applications in preventive and restorative dentistry.

## Figures and Tables

**Figure 1 microorganisms-08-01410-f001:**
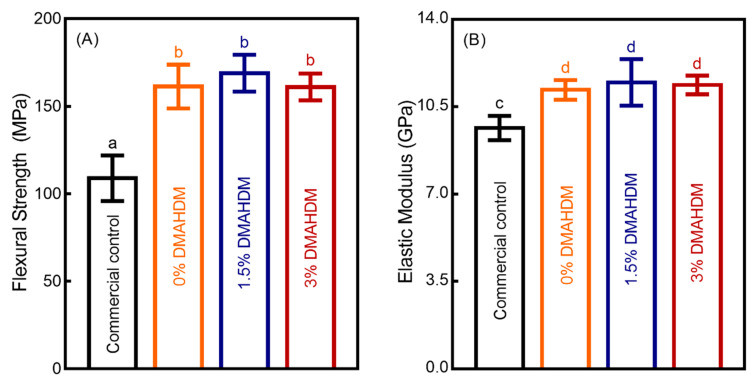
Mechanical properties of composites. (**A**) Flexural strength, and (**B**) elastic modulus (mean ± SD; *n* = 6). Values with disparate letters (a, b, c, d) indicate data are significantly different (*p* < 0.05).

**Figure 2 microorganisms-08-01410-f002:**
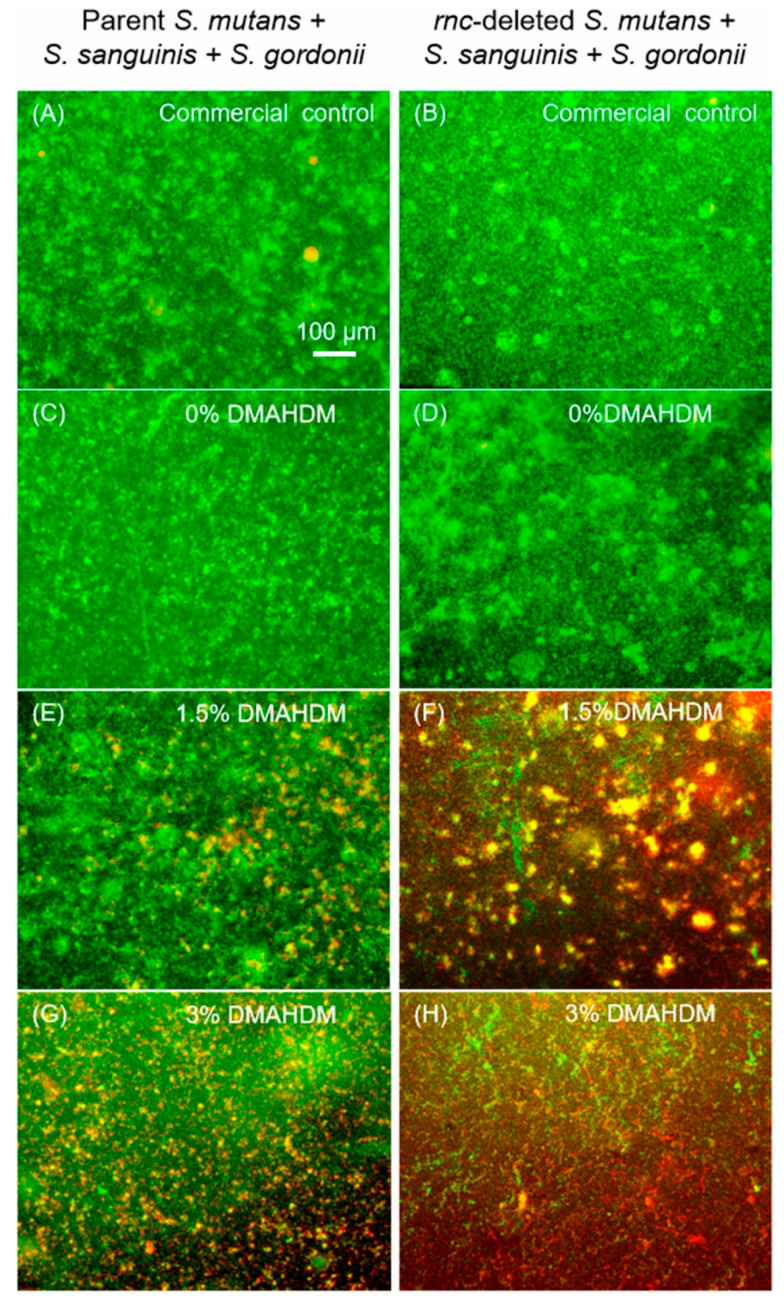
Representative live/dead staining images. Live/dead staining of 2-day Sm + Ss + Sg biofilms on composite: (**A**,**B**) Commercial control, (**C**,**D**) 0% dimethylaminohexadecyl methacrylate (DMAHDM), (**E**,**F**) 1.5% DMAHDM, (**G**,**H**) 3% DMAHDM. All images had the same magnification as (**A**). Live bacteria were stained green, and dead bacteria were stained red. When live and dead bacteria were close to each other or on the top of each other, resulting in yellowish or orange colors.

**Figure 3 microorganisms-08-01410-f003:**
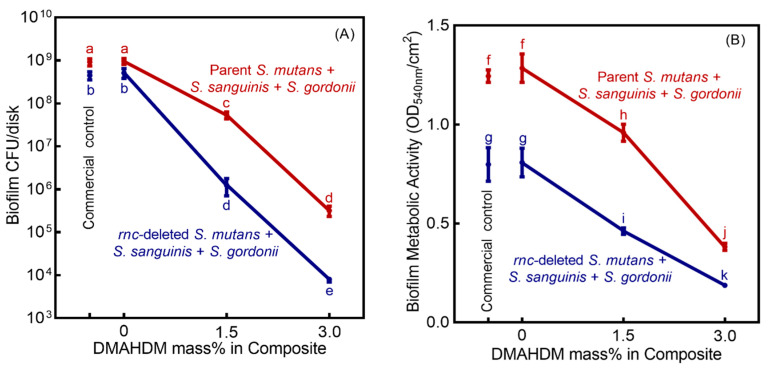
Antimicrobial effects of *rnc* knockout for *S. mutans* plus DMAHDM composite (mean ± SD; *n* = 6). (**A**) Colony-forming units (CFUs), and (**B**) 3-(4,5-dimethylthiazol-2-yl)-2,5-diphenyl tetrazolium bromide (MTT) metabolic activity of Sm + Ss + Sg biofilms on composites. Values with disparate letters (a, b, c, d, e, f, g, h, i, j, k) indicate data are significantly different (*p* < 0.05).

**Figure 4 microorganisms-08-01410-f004:**
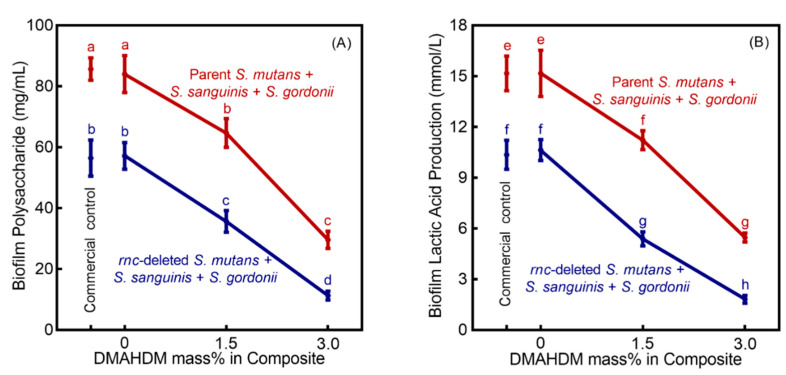
Inhibition effects of *rnc* knockout for *S. mutans* plus DMAHDM composite against cariogenic activities of Sm + Ss + Sg biofilms (mean ± SD; *n* = 6). (**A**) Polysaccharide production by Sm + Ss + Sg biofilms on composites. (**B**) Lactic acid secretion by Sm + Ss + Sg biofilms. Values with disparate letters (a, b, c, d, e, f, g, h) indicate data are significantly different (*p* < 0.05).

**Figure 5 microorganisms-08-01410-f005:**
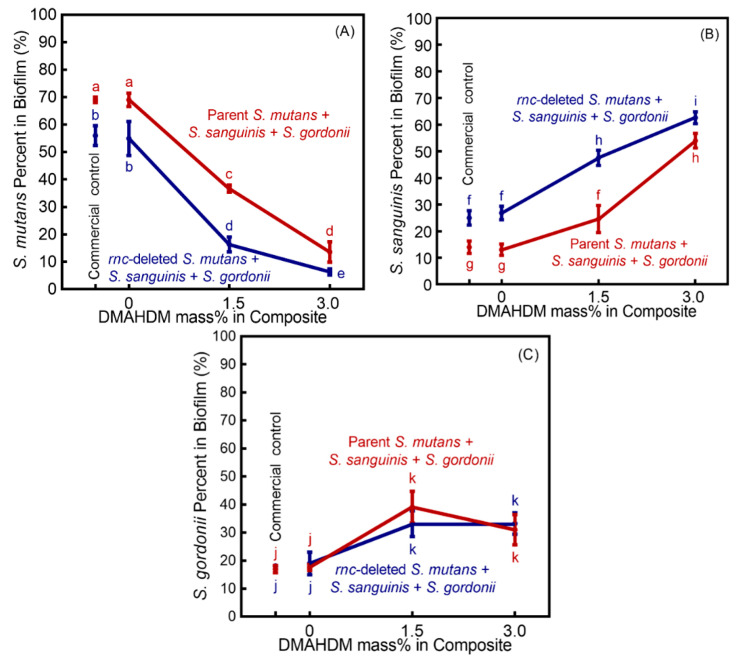
Bacterial species shift in Sm + Ss + Sg biofilm grown for 2 days on composites (mean ± SD; *n* = 3). (**A**) cariogenic *S. mutans* proportion, (**B**) non-cariogenic *S. sanguinis* proportion, and (**C**) non-cariogenic *S. gordonii* proportion. Cariogenic *S. mutans* had overwhelming proportions in commercial control and 0% DMAHDM group. However, with increasing DMAHDM mass fraction, the proportion of cariogenic *S. mutans* decreased sharply, whereas noncariogenic *S. sanguinis* and *S. gordonii* achieved a predominant proportion in the biofilms. Moreover, the effects of *rnc* gene-knockout for *S. mutans* plus DMAHDM composite induce less *S. mutans* ratio. Values with disparate letters (a, b, c, d, e, f, g, h, i, j, k) indicate data are significantly different (*p* < 0.05).

**Figure 6 microorganisms-08-01410-f006:**
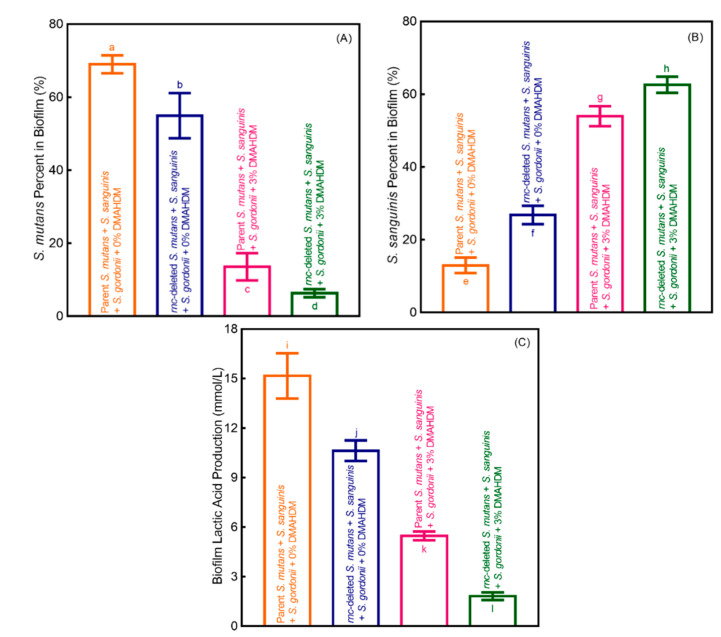
Effects of *rnc*-knockout alone, DMAHDM composite alone, the combination of *rnc*-knockout plus DMAHDM composite, on biofilm species shift and acid secretion. (**A**) Cariogenic *S. mutans* proportion, (**B**) noncariogenic *S. sanguinis* proportion, and (**C**) lactic acid secretion of Sm + Ss + Sg biofilm grown for 2 days on composites (mean ± SD; *n* = 6). In each plot, values with disparate letters (a, b, c, d, e, f, g, h, i, j, k, l) indicate data are significantly different (*p* < 0.05).

**Table 1 microorganisms-08-01410-t001:** Bacterial strains used in this study.

Strain	Description	Source or Reference
Parent *S. mutans*	*S. mutans* wild type UA159;	^a^ ATCC 700610
*rnc* knockout *S. mutans*	Smurnc; The parent *S. mutans* UA159 with inframe replacement by an erythromycin cassette	[33,35]
*S. sanguinis*	*S. sanguinis* wild type	^a^ ATCC10556
*S. gordonii*	*S. gordonii* wild type	^a^ ATCC10558

^a^ ATCC: American Type Culture Collection.

**Table 2 microorganisms-08-01410-t002:** Primers and probes used in TaqMan qPCR.

Primer/Probes	Nucleotide Sequence	Annealing Temperature (°C)	Amplicon Size (bp)	Reference
Primers in TaqMan qPCR				
*S. mutans-*f	5′ GCCTACAGCTCAGAGATGCTATTCT 3′	59	114	[45]
*S. mutans-*r	5′ GCCATACACCACTCATGAATTGA 3′			
*S. sanguinis-*f	5′ GAGCGGATGGCCAATTATATCT 3′	59	75	[44]
*S. sanguinis-*r	5′ CCGGATGATGTCGGCAATA 3′			
*S. gordonii-*f	5′ GGTGTTGTTTGACCCGTTCAG 3′	59	96	[46]
*S. gordonii-*r	5′ AGTCCATCCCACGAGCACAG 3′			
Probes in TaqMan qPCR				
*S. mutans*	5′ FAM-TGGAAATGACGGTCGCCGTTATGAA-TAMRA 3′			[45]
*S. sanguinis*	5′ FAM-TGTTCGGGCTCATGATA-Eclipse 3′			[43]
*S. gordonii*	5′ FAM-AACCTTGACCCGCTCATTACCAGCTAGTATG-TAMRA 3′			[46]

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
