# Peer review of "Regulating Oral Biofilm from Cariogenic State to Non-Cariogenic State via Novel Combination of Bioactive Therapeutic Composite and Gene-Knockout"

_microorganisms, 2020, doi:10.3390/microorganisms8091410_

Round 1

Reviewer 1 Report

The manuscript is a thorough piece of solid work, performed with care and scientific rigour and, further, very well written and understandable. I have just three small comments to it:

  1. The quality of the figures in my copy was not so good due to, perhaps, a poor resolution. Could this be solved?
  2. I would like very much to see in the Introduction, a comment (if any) on the possible toxicity of the polymers used; it is only mentioned in line 22 of the Discussion paragraph.
  3. It would be interesting to discuss the results presented by the group of A. Mira on Streptococcus dentisani (Odontology. 2020 Apr;108(2):180-187; doi: 10.1007/s10266-019-00458-y).

Author Response

The manuscript is a thorough piece of solid work, performed with care and scientific rigour and, further, very well written and understandable.

Thank you very much for the excellent review.

I have just three small comments to it:

Thank you very much for your comments.

  1. The quality of the figures in my copy was not so good due to, perhaps, a poor resolution. Could this be solved?

Excellent point. As you suggested, we have improved the resolution for all the figures in the revised manuscript.

  1. I would like very much to see in the Introduction, a comment (if any) on the possible toxicity of the polymers used; it is only mentioned in line 22 of the Discussion paragraph.

Excellent point. We have added the following statement in the Materials and Methods Section: “In addition, these QAMs with varying alkyl chain lengths had an acceptable low cytotoxicity against human gingival fibroblasts and odontoblast-like cells, matching the low cytotoxicity of commercial dental monomers in clinical use [31,32]. DMAHDM with an alkyl chain length of 16 was incorporated into resins, and presented a strong antimicrobial ability with a low level of cytotoxicity [30,32].”  

  1. It would be interesting to discuss the results presented by the group of A. Mira on Streptococcus dentisani (Odontology. 2020 Apr;108(2):180-187; doi: 10.1007/s10266-019-00458-y).

Excellent point. As your suggestion, we have added the following statement near the end of the Discussion Section: “A previous study [64] reported a unique method of probiotics bacterial colonization to prevent tooth decay. In addition, further study is still needed to investigate the S. mutans rnc gene-knockout effect in an oral biofilm model in vivo with animal experiments.”

Thank you very much for an excellent review.  We look forward to hearing from you.

Best Regards,

Hockin Xu

Professor, Director, Biomaterials & Tissue Engineering Division

Reviewer 2 Report

Dear authors,

It's an interesting and well written study.

Some remarks: 

P2 line 10-14 It will be interesting to explain that even healthy people have a cariogenic biofilm (doi: 10.3390/microorganisms7090319, DOI: 10.1371/journal.pone.0185804)

P4 line 20 « The use of all the bacterial species was approved by the *** Institutional Review Board. Parent S. mutans strain UA159 (ATCC 700610), and the rnc-knockout strain were provided by ***[28, 29]. “ the use of “***” is not clear please modify

P6 line 3 please develop the « 2.9 DNA isolation and TaqMan qPCR assay “ What technique did you used to extract the DNA? Give the temperature of hybridization of primers? Give the length of the amplicons? How did you verify the specificity of the primers?

In the discussion section, please discuss the fact that actually the role of S. mutans in carious lesion is discussed. Some studies demonstrated a real association between S. mutans and the carious lesion, whereas others revealed no clear association (PMCID: PMC3825179). In some studies, caries lesions occurred without the presence of S. mutans (doi: 10.1371/journal.pone.0047722., doi: 10.1177/0022034515609554….)

Author Response

It's an interesting and well written study.

Thank you very much for your comments and recommendations.

Some remarks:

1. P2 line 10-14 It will be interesting to explain that even healthy people have a cariogenic biofilm (doi: 10.3390/microorganisms7090319, DOI: 10.1371/journal.pone.0185804)

Excellent point. We have added the following sentences in the first paragraph of Introduction Section: “To date, over 700 phylotypes have been found in the oral cavity of healthy people and people with diseases [1]. Even healthy people can present cariogenic bacteria [2,3]. It is important to maintain the balance of oral microbiome for oral and body health [4,5].”......“Dental caries is associated with dysbiosis of the tooth-colonizing microbiota, with the species types in the dental plaque biofilm shifting from a healthy composition toward a cariogenic composition [8,9].”

2. P4 line 20 « The use of all the bacterial species was approved by the *** Institutional Review Board. Parent S. mutans strain UA159 (ATCC 700610), and the rnc-knockout strain were provided by ***[28, 29]. “ the use of “***” is not clear please modify

Excellent point. We have changed the following sentences in the Materials and Methods Section: “The use of all the bacterial species was approved by the University of Maryland Baltimore Institutional Review Board. Parent S. mutans strain UA159 (ATCC 700610) and the rnc-knockout strain were provided by West China School of Stomatology [35,36]. Prior to the experiments, the accuracy of the strains was verified using polymerase chain reaction (PCR) and sequencing [35].”

3. P6 line 3 please develop the « 2.9 DNA isolation and TaqMan qPCR assay “What technique did you used to extract the DNA? Give the temperature of hybridization of primers? Give the length of the amplicons? How did you verify the specificity of the primers?

Excellent point. DNA was isolated and purified with the Genomic DNA Extraction Kit (TaKaRa Biotechnology Co., Ltd, Japan) following the manufacturer’s protocol. TaqMan qPCR was used to monitor bacteria composition shifts in the biofilms at 48h.  Conventional PCR was used to confirm the specificity of the primers. Following your suggestion, we have added the description in the Materials and Methods Section: “TaqMan real-time PCR was used to quantify the bacteria composition shifts at 2-day biofilms using a Premix Ex Taq (Probe qPCR). The species-specific primers and probes used for S. mutans, S. sanguinis and S. gordonii were designed following the methods described previously (Table 2) [8]. Conventional PCR was used to confirm the specificity of the primers. Biomass samples were collected, and total bacterial DNA isolation and purification were performed with the Genomic DNA Extraction Kit (TaKaRa Biotechnology Co., Ltd, Japan) following the manufacturer’s protocol [8]. NanoDrop 2000 spectrophotometer (Thermo Scientific, Waltham, MA, USA) was used to determine the purity and concentration of the DNA.”  In addition, we have added the annealing temperature and Amplicon size in Table 2.

4. In the discussion section, please discuss the fact that actually the role of S. mutans in carious lesion is discussed. Some studies demonstrated a real association between S. mutans and the carious lesion, whereas others revealed no clear association (PMCID: PMC3825179). In some studies, caries lesions occurred without the presence of S. mutans (doi: 10.1371/journal.pone.0047722., doi: 10.1177/0022034515609554….)

Excellent point. Following your suggested, we have added the following sentences in the Discussion Section: “As a key modulator, S. mutans is the primary EPS producer in the oral cavity and is responsible for dental caries [14,15]. The acidification of EPS-rich matrix favors the growth of the cariogenic pathogens, promotes demineralization of the enamel apatite on the teeth, and leads to the onset of dental caries [52,53]. Although S. mutans is an important player in caries pathogenesis, its role in caries is not etiological [54]. The correlation between caries experience with elevated numbers of S. mutans has been controversial and not well-established [54-56]. Some studies demonstrated a real association between S. mutans and the carious lesion, whereas other studies revealed no clear association [57]. In some studies, caries lesions occurred without the presence of S. mutans [58].” .….. “Oral streptococci account for 20% of all supragingival microorganisms in the oral biofilm, but they account for nearly 80% of the initial colonizers during the early stages of biofilm formation [8,56]. Furthermore, oral streptococci have been associated with the onset and progression of carious lesions [8,56].”

Thank you very much for an excellent review.  We look forward to hearing from you.

Best Regards,

Hockin Xu

Professor, Director, Biomaterials & Tissue Engineering Division